# A Computational Model for Analysing the Dry Rolling/Sliding Wear Behaviour of Polymer Gears Made of POM

**DOI:** 10.3390/polym16081073

**Published:** 2024-04-12

**Authors:** Aljaž Ignatijev, Matej Borovinšek, Srečko Glodež

**Affiliations:** Faculty of Mechanical Engineering, University of Maribor, Smetanova 17, 2000 Maribor, Slovenia; aljaz.ignatijev@student.um.si (A.I.); matej.borovinsek@um.si (M.B.)

**Keywords:** polymer gears, rolling/sliding contact, wear, computational modelling

## Abstract

This study presents a computational model to determine the wear behaviour of polymer gears. Using PrePoMax finite element numerical calculation software, a proposed computational model was built to predict dry rolling/sliding wear behaviour based on Archard’s wear model. This allows the calculation of the wear depth in each loading cycle with constant mesh updating using the finite element method. The developed computational model has been evaluated on a spur gear pair, where the pinion made of POM was meshed with a support gear made of steel. The computational results obtained were compared with the analytical results according to the VDI 2736 guidelines. Based on this comparison, it was concluded that the proposed computational model could be used to simulate the wear behaviour of contacting mechanical elements like gears, bearings, etc. The main advantage of the model, if compared to the standardised procedure according to the VDI 2736 guidelines, is the geometry updating after a chosen number of loading cycles, which enables a more accurate prediction of wear behaviour under rolling/sliding loading conditions.

## 1. Introduction

Polymer gears are used widely in many engineering applications, such as office appliances, mechatronic devices, household facilities, computer and laboratory equipment, medical instruments, etc. [1,2,3,4,5,6,7]. In these applications, the contact of two mechanical elements (i.e., gear flanks) is concentrated on a small contact area, resulting in higher contact stresses, which often exceed the yield stress of the material. Due to repeated contact loading during the exploitation, local cyclic plastic deformations may lead to surface damage, usually termed surface wear [8,9,10,11]. 

Many wear models relating to rolling/sliding contacting mechanical elements have been proposed in recent years. However, the Archard’s model [12] is used widely in engineering applications. The model requires information about the coefficient of wear, relative sliding distance and load, and has been applied to gear wear prediction by many researchers, who have derived equations that connect the wear volume or wear depth with the operating conditions and material properties of the gear tooth. Flodin et al. [13,14] derived a load distribution equation to determine the transferred load between the spur and helical gear pairs, using the Herztzian contact theory to calculate the contact stress and the single point tracking method for relative sliding distance. Later, this model was extended to analyse the wear behaviour of double helical gears [15], worm gears [16], internal gears [17], hypoid bevel gears [18,19,20] and planetary gears [21,22].

Discretising the Archard model allows for a more detailed analysis of the wear behaviour of materials, especially where wear varies across the materials’ surfaces. Põdra et al. [23] used the finite element method for simulating sliding wear between the rotation disk and the pin. For rolling/sliding applications, such as gears, Lundvall et al. [24,25] derived time increment equations to predict wear using the nonsmooth Newton method. Employing the finite element method opens more possibilities to improve accuracy, which is why Bajpai et al. [26] combined the finite element method and the Archard model, using the iterative method to consider contact changes due to wear. Because mechanical elements like gears, bearings, etc., are dynamically loaded components, it is also useful to consider time-varying sliding distance and time-varying contact load in the wear analysis to make results more accurate [27,28,29,30,31,32].

In recent years, polymer gears have become used more widely, due to their positive properties, such as lubrication-free operation, corrosion resistance, lightweight, arbitrary geometry and low cost of manufacture. They are found mostly in gears within small appliances, where they can be combined with a variety of other materials [33]. As they are under cyclic load, they are dynamically loaded components. As the gears roll and slide between the tooth flanks of the gears, erosion and wear occur on the surfaces over time, which can be determined with experiments, analytically or numerically. The analytical calculation of polymer gears can be found in the VDI 2736 Standard (Association of German Engineers) [34]. This covers the basic calculation and the presentation of the properties of polymer gears. Among those properties, the wear depth of the gears can be calculated using this Standard. Although this Standard can be used to obtain certain values, it is limited to constant operating conditions, which, unlike gears from metallic materials, can change rapidly due to their poorer mechanical properties. As mentioned, the finite element method provides more accurate results and allows rapid changes of material properties and operating conditions. Lin et al. [35] investigated the interaction between contact loads and tooth wear in POM (poly-oxy-methylene) and nylon tooth pairs, using the finite element method to analyse tooth stiffness and a quasi-linear iterative method to calculate wear depth. The results show that peak load increases as the number of cycles reaches a threshold limit.

In the framework of the presented study, a computational model was built using the open-source finite element numerical calculation software PrePoMax 1.4.0 to predict wear behaviour based on Archard’s wear model, allowing for the calculation of wear depth in each loading cycle with constant mesh updating. Polyoxymethylene (POM) was used for the evaluation of the computational model due to its frequent use in gearing applications, and it represents a relevant material for the study of the dry rolling/sliding wear behaviour of polymer gears. POM was recognised for its high mechanical strength and stiffness, good sliding properties and wear resistance. It offers excellent dimensional stability and is easy to machine.

The computational results obtained in the framework of this study were compared with the analytical results according to the VDI 2736 guidelines. Some of the computational studies described above have already used FEM with an implementation of the Archard model and a geometry updating method. More recent studies have also considered dynamic loading, but all of these studies are based on the wear of metallic materials, which are more resistant against wear if compared to polymeric materials. At the same time, polymer gears experience additional deformation due to their lower stiffness. The novelty of this work lies in the development of a computational model that updates polymer gear geometry after a certain number of loading cycles. The main objective of this study’s approach to wear prediction of polymer gears is to make the simulation of the wear process more dynamic and realistic. This approach provides a more accurate determination of the service life of these components concerning wear. In that respect, the proposed computational model could represent an effective tool for analysing the wear behaviour of rolling/sliding mechanical elements made of polymeric materials.

## 2. Materials and Methods

### 2.1. Computational Modelling

In this study, the PrePoMax open-source pre- and post-processor for the finite element method (FEM) [36] was used to analyse the wear behaviour of rolling/sliding contacting mechanical elements (i.e., mating gear flanks). The solver used by PrePoMax is the open-source CalculiX FEM solver [37], which does not support the wear analysis by itself, so the wear analysis was implemented into the PrePoMax. The linear Archard wear model [38] for predicting the volume of worn material was chosen for this purpose. The model was originally developed based on the experimental results of the cyclic wear of metal specimens under dry contact conditions. The model is represented by the following equation:(1)$$V=k\cdot \frac{F}{H}\cdot s$$
where V is the volume of worn material, k is the dimensionless wear coefficient, F is the normal contact force, H is the surface hardness in contact and s represents the sliding distance. To account for changes in the geometry of the contact surfaces during wear cycles, the aforementioned model, which neglects the geometry change, can be divided into multiple steps when used in numerical analyses using the FEM. In a multi-step numerical procedure, where the depth of wear needs to be calculated for each wear cycle [39], Equation (1) can be expressed as follows:(2)$$h_{i}=\int \frac{k}{H}\cdot p_{i}\cdot ds$$
where $h_{i}$ represents the depth of wear in the *i*-th wear cycle, $p_{i}$ denotes the normal contact pressure, and ds represents the differential sliding distance. The total wear depth, h, over n wear cycles is then computed by Equation (3), summing up the wear depths of individual cycles. Such an approach enables the wear analysis to be independent of the model geometry, boundary conditions and loading.
(3)$$h=\sum_{i=1}^{n}h_{i}$$

The multi-step method implemented in PrePoMax is based on the following assumptions: (i) The entire wear process can be divided into multiple repetitive wear cycles. (ii) Wear occurs only on the softer surface in the contact pair (the slave surface). (iii) The change in contact surface geometry during one wear cycle is negligible. (iv) The dimensionless wear coefficient is constant.

A finite element model specific to that cycle needs to be prepared for each wear cycle. The model must capture the evolving contact conditions accurately during the cycle, requiring a nonlinear contact analysis solved in multiple increments. For an accurate result, the number of increments must be large enough to result in a smooth movement of the contact pressure over the finite element mesh of the contact surfaces.

The wear calculation for the cycle begins after the nonlinear contact analysis is completed. All wear parameters are computed only at those slave surface finite element nodes that were in active contact with the master surface during the wear cycle. The normal contact pressure, $p_{i,j}$, at node j and the nodal relative sliding distance, $∆s_{i,j}$, are determined from the resulting contact fields. The wear depth at node $h_{i,j}$ is then computed using the expression:(4)$$h_{i,j}=\frac{k_{j}}{H_{j}}\cdot p_{i,j}\cdot ∆s_{i,j}$$
where the material properties *kj* and *Hj* are taken at node *j*. The processing of each wear cycle concludes with the calculation of the change in the contact surface geometry due to wear, determined by the wear displacements at the nodes calculated from the wear depth. The node wear displacements due to wear in each cycle, ${\overset{⃑}{h}}_{i,j}$, are calculated by a scalar product:(5)$${\overset{⃑}{h}}_{i,j}=-h_{i,j}\cdot {\overset{⃑}{n}}_{j}$$
where ${\overset{⃑}{n}}_{j}$ represents the node’s outward normal vector. The node’s outward normal vector is determined as an average value of the normal vectors of the FEs sharing node j (Figure 1). At the start of the next wear cycle, the wear displacements computed in the previous cycle are used to update the mesh. Thus, the simulation of the new wear cycle is carried out on a mesh where the wear is considered from the previous cycles. 

Applying the wear displacements to the mesh changes only the position of the surface nodes, and thus deforms the shape of the finite elements, making all surface elements on the wear surface a little thinner. If the accumulated wear displacement is large enough, this might lead to unusable finite element shapes. This is especially problematic when a dense mesh is used on the contact surfaces to obtain an accurate contact pressure field, since even a small wear displacement will deform a small finite element beyond being usable. To overcome this problem, local or global remeshing of the domain can be applied, or a method for mesh redistribution can be used. In the PrePoMax software, a boundary displacement method (BDM) [40] for mesh redistribution was introduced. This is a two-step procedure, where the surface nodes are first moved, and then new positions are computed for the internal nodes. From many different approaches to the BDM, the positions of the internal nodes were determined based on the solution of the linear elastic solid body problem. This was the most convenient approach, since it can be solved with the FEM method on the existing mesh. The method was integrated into PrePoMax by an additional BDM simulation step that is added after each wear cycle. In this simulation step, the mesh from the previous wear step is taken, uniform elastic properties are assigned to all elements and a prescribed boundary condition is applied to all nodes on the surface. In the nodes where the wear occurs, the value of the prescribed displacement vector is equal to the wear displacement, ${\overset{⃑}{h}}_{i,j}$, and equal to $\overset{⃑}{0}$ for all other nodes. After the BDM simulation step is completed, the computed nodal displacements for all nodes from the BDM step are used to deform the mesh before the next wear step is started.

Running multiple wear steps one after the other and deforming the mesh after each step can lead to the occurrence of high-frequency oscillations (roughness) of the mesh (see Figure 1), which results in an irregular contact pressure field. This causes an irregular wear displacement field, which additionally enhances the roughness of the contact surfaces and renders the results of the wear analysis unusable. A mesh smoothing algorithm on the wear displacement field was introduced to remove the high-frequency oscillations from the mesh. The smoothing algorithm is based on the Laplacian mesh smoothing procedure [41], where the new position of node ${\overset{⃑}{x}}_{j}$ is computed from the positions of its neighbouring nodes, ${\overset{⃑}{x}}_{k}$, as:(6)$${\overset{⃑}{x}}_{j}=\frac{1}{m}\sum_{k=1}^{m}{\overset{⃑}{x}}_{k}$$
where m equals the number of node neighbours. Mesh smoothing is applied after the wear displacement field is computed and before the BDM step is prepared. The effect of the smoothing step can be increased by running the smoothing algorithm multiple times, which can be defined by the user. All these features make wear analyses applicable and robust for a large variety of problems, and the user can prepare the wear analysis without additional coding using only the built-in user interface. This makes the implementation usable for many engineers since all these features are already available in the latest released version of PrePoMax v1.4.0, available online [42].

### 2.2. Geometrical and Material Parameters of the Analysed Gear Pair

The developed computational model has been evaluated on the spur gear pair, where the pinion made of POM has meshed with a support gear made of steel. The basic geometrical and material parameters of the analysed gear pair are presented in Table 1.

### 2.3. FE Model of Gear Pair

The geometry of the gear pair is shown in Figure 2, where the pinion is shown in grey and the support gear is shown in green. Only one tooth of the gear and three teeth of the pinion are presented here. This was due to the smaller stiffness of the pinion material (POM) if compared to the gear material (steel). The analysis of one load cycle was split into three simulation steps, as shown in Figure 3: a static step (contact initialisation), a wear step (computation of wear) and a BDM step (mesh redistribution). In the simplified model presented in Figure 2, the reference points RP1 and RP2 have been defined in both axes of the gear pair. Here, RP1 was connected rigidly to the gear shaft, with the surfaces of the gear marked in red. Additionally, at this reference point a torque of *T* = 16 Nm was applied acting to the pinion tooth, and a boundary condition was imposed, which allowed the pinion to move only about its axis. The point RP2 was connected rigidly to the pinion surfaces marked in blue. At this reference point a boundary condition was prescribed first in the static step, which prevented displacement and rotation in all directions of the coordinate system. As sliding was required for the wear calculation procedure, the boundary condition in RP1 in the second step (Wear step) required a rotation around the pinion axis of ρ = 0.285 rad. The angle of rotation was prescribed in such a way that the gear only reached the position where the gear started to return to the already worn surface.

The computational model implemented into the PrePoMax software (see Section 2.1) was then used to analyse the wear behaviour of mating gear flanks. The numerical simulation of gear meshing consisted of three separate simulation steps, which followed one after the other. The simulation was then repeated to consider the selected number of load cycles. 

Archard’s model implemented in the PrePoMax software is limited to a 3D wear calculation, as it calculates the volume of wear. However, as the pinion and the gear analysed in this study do not vary in thickness, it would be reasonable to simplify the model to 2D, but this was not possible. Therefore, a reduced thickness of 0.25 mm was assumed to reduce the computational time, and, consequently, the load (torque) was also reduced proportionally. Furthermore, an additional boundary condition was added to both gears, to prevent the gears from moving in the thickness direction. To approximate the 2D model best, a finite element mesh with prismatic finite elements was created using the boundary layer function, as shown in Figure 4. The overall model was meshed with a global mesh size of 1 mm. However, for better accuracy the model around the contact was meshed with a local mesh size of 0.05 mm, which resulted in 3300 parabolic prismatic finite elements. The model used a linear overclosure–pressure contact relation, with the stiffness *K* = 10^5^ N/mm^3^ and coefficient of friction *μ* = 0.25 mm.

Two computational models were considered in the numerical simulation. In the first model (Model 1), a wear calculation was determined after 1,000,000 load cycles. First, for comparison with VDI 2736, the 1,000,000 load cycles were calculated with the number 1,000,000 as a cycle increment, which means that the geometry was not updated during the calculation. The purpose of this computational model was to compare the results with the VDI 2736 calculation, which did not consider the change in geometry. In the second model (Model 2), 1,000,000 load cycles were reused. However, compared to the first model, the number of cycles was divided into ten steps, meaning that the geometry was updated after every 100,000 load cycles. The wear results were thus transferred directly to the finite element mesh, and the wear was recalculated on the updated geometry.

The main assumptions of the computational model are that only one gear tooth is in contact during meshing, that only one gear tooth in a pair gets worn out and that the wear is linear and contact pressure independent. Only one gear tooth in contact was selected to compare the numerical results with the analytical results, but additional gears could be added to the model. The wear on the steel gear tooth was not computed during the analysis, since it was assumed that the amount of wear in steel is negligible and does not change the tooth geometry enough to influence the meshing. The wear model was regarded as linear in order to greatly decrease the computational time, since one simulation cycle can represent multiple wear cycles in a linear model. The wear coefficient was assumed to be contact pressure independent in order to compare it to the analytical approach to verify the numerical model.

### 2.4. Analytical Approach According to VDI 2736

Following the calculation according to VDI 2736, a wear behaviour can be obtained for the tooth flank profile of the polymer gear. The calculation is position-dependent, as the wear is different at each point due to the involute shape of the tooth. Here, the wear starts to decrease from the tip of the tooth up to the pitch point. Due to the involute shape of the tooth, sliding dominates at the starting engagement point, then the tooth rolls at the pitch circle, and later, towards the root of the tooth, sliding increases gradually again, and rolling decreases. Therefore, in the pitch point region, it can be observed that the wear is theoretically zero. Considering this fact, the local wear of the gear flank can be obtained as follows:(7)$$W_{local}=\frac{F_{n}}{b_{w}}\cdot N_{L}\cdot \zeta \cdot k_{w}$$
where *b_w_* is a common face width, *N_L_* is the number of load cycles, *ζ* is the specific local sliding of gear flanks, *k_w_* is the wear coefficient and *F_n_* is the normal force acting on the tooth flank.
(8)$$F_{n}=\frac{2\cdot T_{d}}{d_{w}\cdot \cos\alpha_{t}}$$
where *T_d_* is the nominal torque, *d_w_* is the pitch circle diameter and α*_t_* is the pressure angle in the transverse section. It follows from Equation (8) that the normal force, *F_n_*, is a constant throughout the engagement line of gear flanks. However, its tangential and radial components change, resulting in a combination of sliding and rolling of the gear flanks. When sliding is involved, a specific local sliding, *ζ*, appears, which should be considered when determining the local wear using Equation (7). Figure 5 shows the sliding conditions in an arbitrary mating point Y before pitch point C. The circular velocities in an arbitrary mating point Y (*v*_y1_ = *r*_y1_·ω_1_ and *v*_y2_ = *r*_y2_·ω_2_), which are acting perpendicular to the lines O_1_Y and O_2_Y, can be distributed in the directions of common normal (*v*_ny1_ and *v*_ny2_) and common tangent (*v*_ty1_ and *v*_ty2_). Here, the normal velocity components must be the same, i.e., *v*_ny1_ = *v*_ny2_. Furthermore, the tangential components *v*_ty1_ and *v*_ty2_ can be obtained from the triangles O_1_T_1_Y and O_2_T_2_Y and the hatched triangles.

The sliding velocities of gear flanks in an arbitrary mating point Y (*v*_gy1_ for a pinion and *v*_gy2_ for a gear) are defined as the difference between the tangential velocity components *v*_ty1_ and *v*_ty2_ (*v*_gy1_ = *v*_ty1_ − *v*_ty2_; *v*_gy2_ = *v*_ty2_ − *v*_ty1_). From triangles O_1_T_1_C and O_2_T_2_C and with consideration that O_1_C = *d*_w1_/2 and O_2_C = *d*_w2_/2, it follows:(9)$$v_{\mathrm{gy}1}=-\;\left(\omega_{1}+\omega_{2}\right)\cdot \bar{\mathrm{YC}}\;;\;v_{\mathrm{gy}2}=+\;\left(\omega_{1}+\omega_{2}\right)\cdot \bar{\mathrm{YC}}$$

Because the sum (ω_1_ + ω_2_) is a constant value, it is clear from Equation (9) that the sliding velocity, *v*_gY_, only depends on the distance of point Y from the pitch point C. When an arbitrary mating point Y coincides with pitch point C, the sliding velocity equals zero.

Based on the sliding velocity *v_gy_*_1_ in an arbitrary mating point Y, the specific local sliding of gear flanks *ζ* is defined as [45]:(10)$$\zeta =\frac{v_{\mathrm{gy}1}}{v_{\mathrm{ty}1}}=\frac{\left(1+\frac{z_{1}}{z_{2}}\right)\bar{YC}}{r_{w1}\cdot sin\alpha_{t}+\bar{YC}}$$

Following the calculation procedure according to VDI 2736, the wear behaviour of gear flanks is position-dependent, due to different sliding velocities along the engagement line of mating gears. It is clear that the sliding velocities and, consequently, the wear of the gear flanks, were the highest at the starting and ending engagement points and then decreased toward pitch point C.

## 3. Results and Discussion

Figure 6 shows the wear of the analysed gear made of POM after 1,000,000 loading cycles considering Model 1, where the geometry change due to wear of the gear flank was not considered in the numerical simulation. The depth of the wear was the highest at the starting and ending engagement points, which corresponds to the theoretical background as described in Section 2.4. Similar conclusions can also be made for Model 2, which considered the geometry change during the numerical simulations (see Figure 7).

Figure 8 shows the comparison between the initial and the worn geometry of the analysed POM gear after 1,000,000 loading cycles (a scaling factor of 3 was considered). The main differences appeared at the starting and ending mating points, where the wear of the gear flank was the highest.

Simulating each load cycle when analysing thousands of load cycles would require too much computational time. At the same time, the geometry of the model does not change significantly during one load cycle, so it is reasonable to increase the number of cycle increments. Figure 9 shows the convergence analysis of incremental cycles. As the cycle increment decreased, the accuracy of the simulation increased. The blue curves represent a larger cycle increment (1,000,000, 500,000 and 200,000) and differ significantly from each other, gradually converging to a dashed curve. The red curve shows a simulation with a cycle increment of 100,000 and the black dashed curve shows a cycle increment of 50,000. As these two curves overlap, we can conclude that the simulation with a cycle increment of 100,000 was optimal in terms of computational time and accuracy.

Figure 10 shows the comparison between the computational results (for Models 1 and 2) and the standardised approach according to VDI 2736. It should be noted that VDI 2736 is based on the wear properties of the materials obtained from a rotating pin-bearing disc test, whereas the final equation for the calculation of the wear depth is based solely on the involute profile of the analysed gear. Based on the obtained computational results using the proposed computational model and analytical results according to VDI 2736, it can be concluded that both approaches provide a similar wear behaviour of the analysed POM gear (the maximum wear appeared at the starting and ending mating points). It is also evident that the computational results according to Model 1 (geometry changes due to wear of the gear flank were not considered) describe the wear behaviour better if compared to VDI 2736. However, it should be pointed out that, in real applications of the gear drive, the geometry of gear flanks changes constantly due to wear, which may affect the subsequent wear phenomenon of the gear flanks significantly. By adding updates to the geometry, we can get closer to more realistic wear values. Furthermore, it is possible to observe the change in wear compared to the model without a geometry update. In Figure 10, the green line shows the path of the wear curve of the Model 2. 

Oscillations of the wear depth value between adjacent nodes have occurred in the computational model. Wear is dependent directly on the contact pressure field, which means that if the contact pressure field is smooth, then the calculated wear field is smooth, and vice versa. A smoothing function was applied, as oscillations occurred during the simulation (the blue curve in Figure 11). A problem occurs when updating the finite element mesh, as large variations occur between adjacent nodes. With each cycle increment, this oscillation increases, causing instability in the simulation. The smoothing function acts as an average filter, averaging the results between adjacent nodes. Figure 10 also shows the difference that occurs at pitch point C. For this reason, the obtained wear curves were corrected with dotted lines in Figure 10. 

The values at pitch point C of Model 2 are slightly higher; this was not only due to smoothing but also to the new tooth geometry, which is no longer ideal because of the tooth deflection. This results in sliding instead of rolling. It is possible to observe a marked change at the root of the tooth. The wear starts to decrease with the number of cycles, also due to the change in geometry. Relief occurs at the root of the tooth, as the maximum pressure is reduced or redistributed.

## 4. Conclusions

In the framework of this study, a comprehensive computational model to analyse the wear behaviour of polymer gears was developed and evaluated on a spur gear pair, where a pinion made of POM was meshed with a support gear made of steel. A proposed computational model was built using the PrePoMax finite element numerical calculation software considering Archard’s wear model to predict the wear behaviour of the analysed contacting surfaces. Based on the obtained computational results and their comparison to the analytical results according to the VDI 2736 guidelines, the following conclusions could be made:The proposed model offers an improved approach to computing the wear between gear flanks and compares it to the analytical approach according to the VDI 2736 guidelines.The use of the boundary displacement method (BDM) in the framework of the PrePoMax software provides information on the geometry change that alters the operating conditions in subsequent load cycles.Due to the contact problem and the formation of contact pressure peaks in the finite element mesh, it is necessary to consider mesh smoothing in the model, to allow a smooth distribution of wear over the surface. This avoids additional convergence problems in the use of BDM, but it does result in the averaging of values in locations where the differences in wear between adjacent element nodes should be larger (such as a pitch point) and is not entirely correct. The validity of the results should, therefore, be checked with a model without smoothing.With the computational model using a multi-step geometry update, more accurate results were obtained, which show a reduced depth of wear at the root of the tooth. In the pitch point region, a non-zero value of the wear depth appeared as the number of cycles increased. The tooth deflection and the new tooth flank geometry have a major impact on this.The main advantage of the model, if compared to the standardised procedure according to the VDI 2736 guidelines, is the geometry updating after a certain number of loading cycles, which enables a more accurate prediction of wear behaviour under rolling/sliding loading conditions.In the future, a comprehensive 3D numerical model will be developed to analyse the interaction of meshing gears. This model aims to compare the results obtained from experimental testing with those derived from numerical simulations.Potential future advancements of the existing Archard’s wear model could include incorporating a contact stress-dependent wear coefficient, as demonstrated in prior research. Currently, the wear coefficient remains constant throughout the analysis. Additionally, the model could be enhanced by establishing a relationship between the wear parameter and wear depth, in order to account for diverse surface improvement techniques.In further work, the proposed computational model could also be extended to consider different operating conditions, such as different gear designs, wear conditions, materials, etc. Furthermore, extensive experimental investigations should be proposed to confirm the computational results.

## Figures and Tables

**Figure 1 polymers-16-01073-f001:**
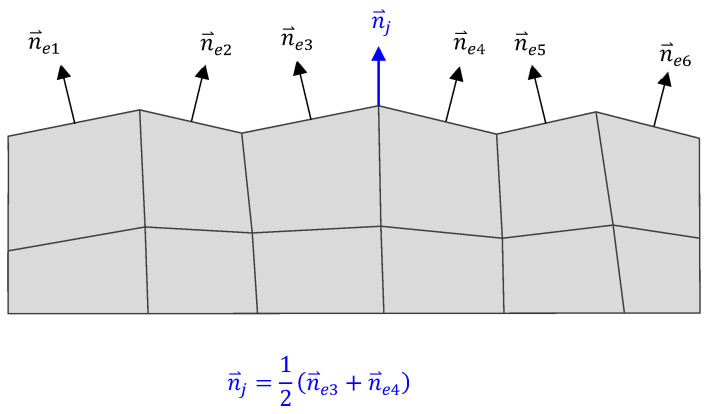
Outward normal vectors of the finite elements and the common node.

**Figure 2 polymers-16-01073-f002:**
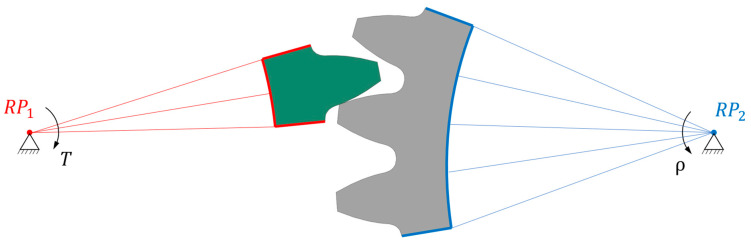
Computational model of gear pair.

**Figure 3 polymers-16-01073-f003:**
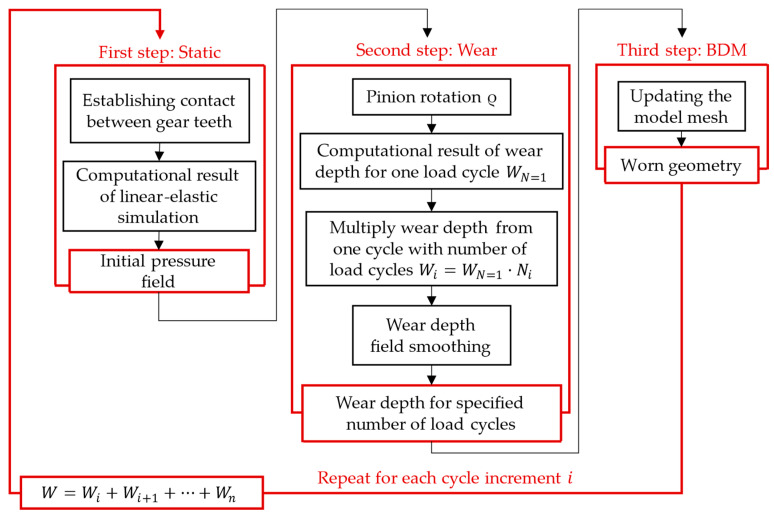
Flowchart of the computational analysis.

**Figure 4 polymers-16-01073-f004:**
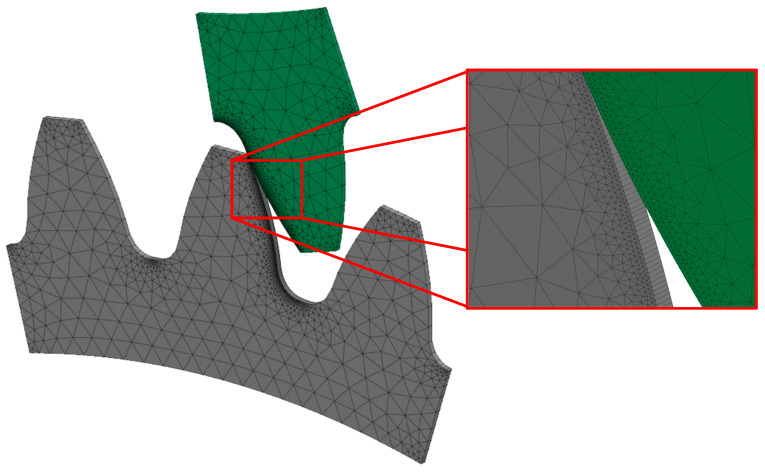
Finite element mesh of a gear pair.

**Figure 5 polymers-16-01073-f005:**
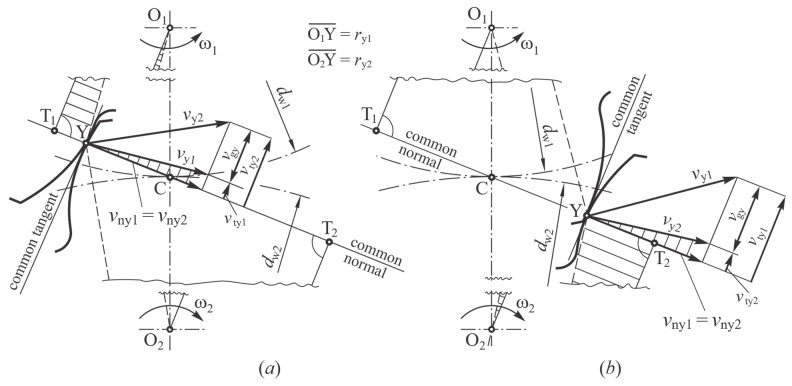
Sliding conditions in an arbitrary mating point Y: (**a**) Before the pitch point, (**b**) After the pitch point.

**Figure 6 polymers-16-01073-f006:**
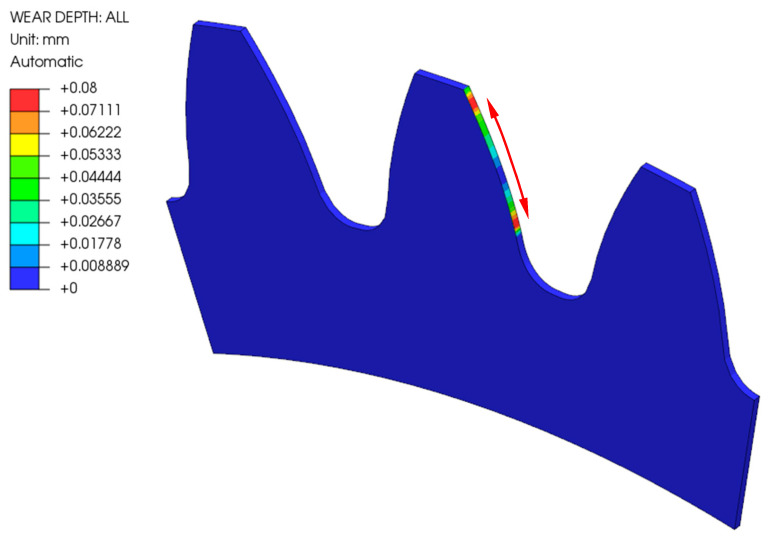
Wear of the POM gear after 1,000,000 cycles (Model 1).

**Figure 7 polymers-16-01073-f007:**
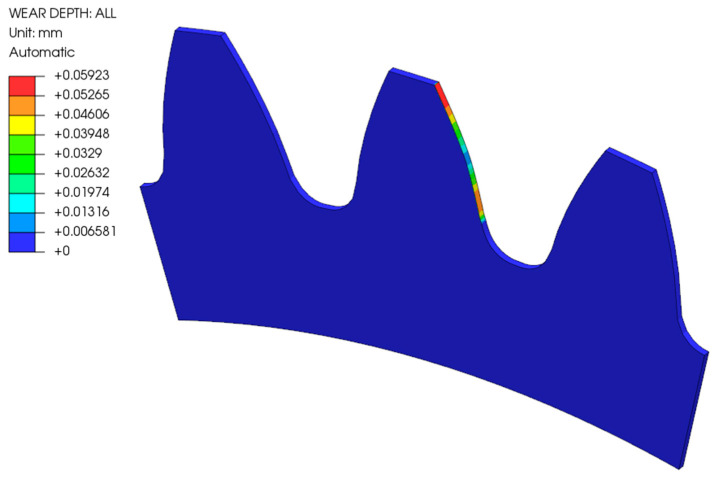
Wear of the POM gear after 1,000,000 cycles (Model 2).

**Figure 8 polymers-16-01073-f008:**
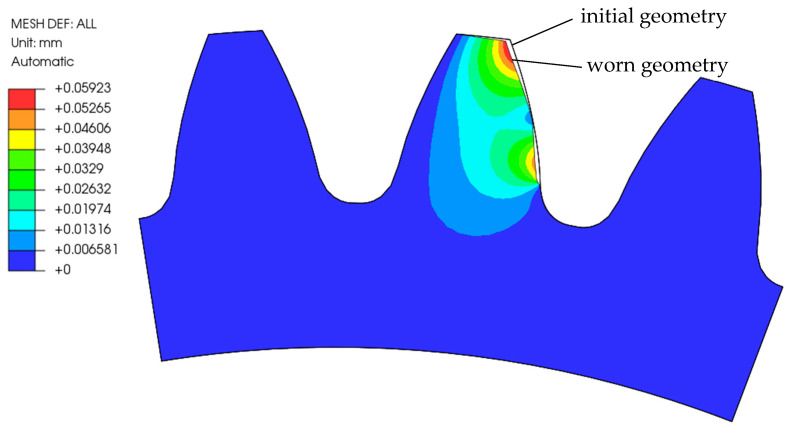
Worn geometry of the POM gear after 1,000,000 cycles (Model 2).

**Figure 9 polymers-16-01073-f009:**
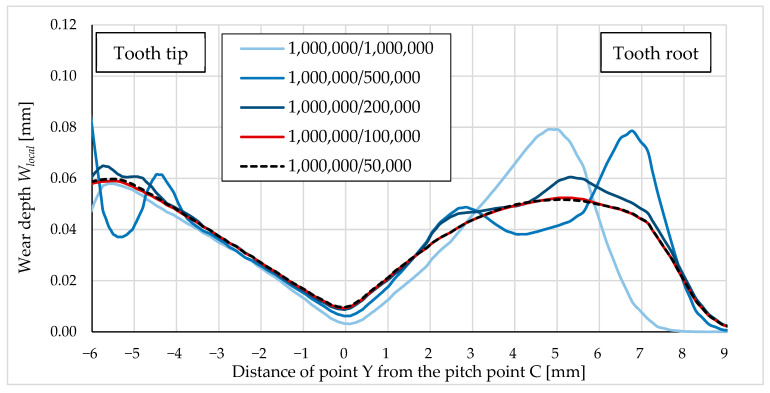
Wear of the POM gear after 1,000,000 cycles (comparison between the computational results for different cycle increments).

**Figure 10 polymers-16-01073-f010:**
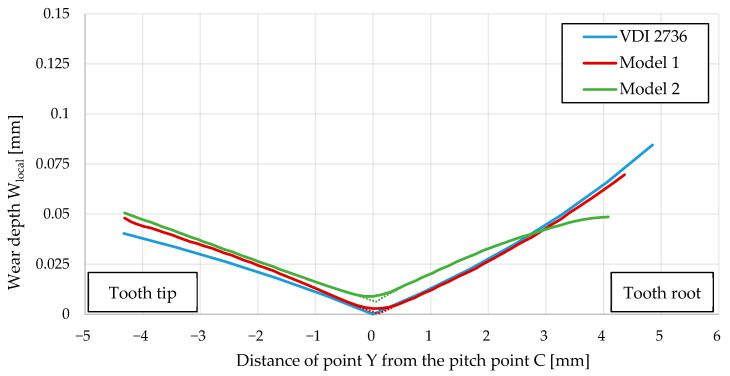
Wear of POM gear after 1,000,000 cycles (comparison between computational results and standardised approach according to VDI 2736).

**Figure 11 polymers-16-01073-f011:**
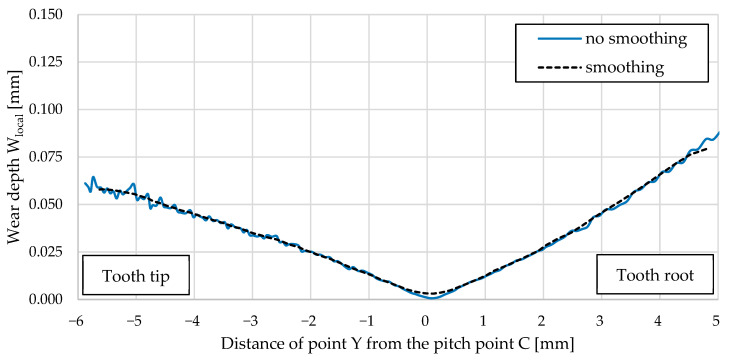
Wear of the POM gear after 1,000,000 cycles (comparison between the results with and without smoothing for a cycle increment of 1,000,000).

**Table 1 polymers-16-01073-t001:** Basic parameters of the analysed gear pair [43].

Parameter	Tested Gear	Supported Gear
Material	POM	Steel (16MnCr5)
Normal module, *m*	2.5 mm	2.5 mm
Pressure angle, α*_n_*	20°	
Helix angle, β	0°	
Number of teeth, *z*	36	36
Tooth width, *b*	14 mm	14 mm
Profile shift coefficient, *x*	0	
Centre distance, *a*	90 mm	
Basic rack profile	ISO 53 [44]	
Young’s modulus, *E*	2600 MPa	210,000 MPa
Poisson’s ratio, ν	0.386	0.280
Lubrication	Dry (not lubricated)	
Wear coefficient, *k*_w_ (VDI 2736)	3.4 × 10^−6^ mm^3^/Nm	

## Data Availability

Data is contained within the article.

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
