# Peer review of "A Computational Model for Analysing the Dry Rolling/Sliding Wear Behaviour of Polymer Gears Made of POM"

_polymers, 2024, doi:10.3390/polym16081073_

Round 1

Reviewer 1 Report

Comments and Suggestions for Authors

The manuscript presents a computational model for analyzing the wear behavior of polymer gears. The topic is quite relevant and addresses an important aspect of gear design. However, some areas need to be improved before acceptance for publication.

- The introduction should provide a clear background for the study, and the objectives should be explicitly stated.

- Additionally, the methodology section should provide the limitations of the computational model and its assumptions.

- Authors need to provide a comprehensive validation and verification of the computational model, including comparisons with experimental data. It will help to establish the accuracy of the proposed model.

- The references should be thoroughly checked for accuracy and consistency. 

Comments on the Quality of English Language

- The manuscript contains several grammatical and typographical errors that need to be corrected.

Author Response

The manuscript presents a computational model for analysing the wear behavior of polymer gears. The topic is quite relevant and addresses an important aspect of gear design. However, some areas need to be improved before acceptance for publication.

Comment #1: The introduction should provide a clear background for the study, and the objectives should be explicitly stated.

Response:

The introduction part was rewritten as suggested.

Comment #2: Additionally, the methodology section should provide the limitations of the computational model and its assumptions.

Response:

The methodology section of the revised manuscript provides the limitations and assumptions as recommended.

Comment #3: Authors need to provide a comprehensive validation and verification of the computational model, including comparisons with experimental data. It will help to establish the accuracy of the proposed model.

Response:

As explained in the Conclusions of the revised manuscript, we plan to do some experimental testing on polymer gear pairs to compare the experimental results directly with the results obtained using the proposed computational model. However, it is important to note that the VDI guidelines to which we are comparing our results are based on some previous experimental studies. As the results are comparable, we can conclude on the adequacy of the proposed computational model.

Comment #4: The references should be thoroughly checked for accuracy and consistency.

Response: The references were thoroughly checked as suggested.

Reviewer 2 Report

Comments and Suggestions for Authors

This paper presents the computational model for analyzing the wear behavior of polymer gears. The main contribution of the paper is combination of FEM method and the Archard model to predict wear depth in each loading cycle with constant mesh updating.

The manuscript shows a lot of promise, but some major issues need to be addressed before it can be published.

1.       This manuscript focused on polymer gear made by POM; thus, please describe the advantages of POM gears and why the authors selected POM as the target material.

2.       If other materials (e.g. PEEK, Nylon…) are used to make polymer gear, how the computational result is obtained? 

3.       The novel of this study could not be found in the introduction part. Please describe the difference between this study and ref. 26~32’s studies in more detail to highlight the relevant contribution to the field.

4.       It is said that “the main advantage of the model is the geometry updating after a certain number of loading cycles, which enables a more accurate prediction of wear behavior under rolling/sliding loading conditions.” However, the wear depth behavior is only significant changed at the tooth root. Please give an explanation to prove your novel finding. And please add more discussion on Fig. 9’s results.

5.       In the conclusion part, authors reported that “Due to the contact problem and the formation of contact pressure peaks in the finite element mesh, it is necessary to consider mesh smoothing in the model to allow a smooth distribution of wear over the surface. This avoids additional convergence problems in the use of BDM, but it does result in the averaging of values in locations where the differences in wear between adjacent element nodes should actually be larger (such as pitch point) and is not entirely correct.” Could authors simulate 3 different areas including the first point of contact, pitch point, and last point of contact with 3 different constant mesh updating?

6.       Is it possible to use both tested gear and supported gears made by POM? The existing Archard's wear model could include incorporating a contact stress-dependent wear coefficient. If authors consider the contact stress-dependent wear, is it necessary to use both tested gear and supported gears made by POM?

Comments on the Quality of English Language

The grammar errors and mistyping words were found. Please check carefully the manuscript again. 

Author Response

General Comment: This paper presents the computational model for analysing the wear behaviour of polymer gears. The main contribution of the paper is combination of FEM method and the Archard model to predict wear depth in each loading cycle with constant mesh updating. The manuscript shows a lot of promise, but some major issues need to be addressed before it can be published.

Comment #1: This manuscript focused on polymer gear made by POM; thus, please describe the advantages of POM gears and why the authors selected POM as the target material.

Response:

In everyday engineering praxis, polymer gears made of POM are recognised for their high mechanical strength and stiffness, good sliding properties and wear resistance. POM offers excellent dimensional stability and is easy to machine. We have chosen POM for the evaluation of the computational model because of its frequent use in gearing applications and because it is a relevant material for the study of the wear behaviour of polymer gears. Some additional explanations are also added in the introduction section of the revised manuscript.

Comment #2: If other materials (e.g. PEEK, Nylon…) are used to make polymer gear, how the computational result is obtained.

Response:

The proposed computational model may also be used to analyse the wear behaviour of polymer gears made of other polymeric materials (e.g. PEEK, Nylon…), considering some material parameters for these polymers. In this study, POM is used as an example of polymer gears.

Comment #3: The novel of this study could not be found in the introduction part. Please describe the difference between this study and ref. 26~32’s studies in more detail to highlight the relevant contribution to the field.

Response:

As described in the introduction and conclusion of the revised manuscript, the proposed computational model allows the determination of wear behaviour in each loading cycle, considering mesh updating. With such a model using a multi-step geometry update, more accurate results can be obtained if compared to the VDI guidelines. Furthermore, the tooth deflection and the new tooth flank geometry have a major impact on the wear behaviour. The references [26] to [32] are focused mainly on applying the Archard model to metal gears, which is quite different if compared to the topic of this study, where polymer gears are analysed.    

Comment #4: It is said that “the main advantage of the model is the geometry updating after a certain number of loading cycles, which enables a more accurate prediction of wear behavior under rolling/sliding loading conditions.” However, the wear depth behavior is only significant changed at the tooth root. Please give an explanation to prove your novel finding. And please add more discussion on Fig. 9’s results.

Response:

Computational results have shown that some wear also appears in the pitch point, which is opposite to the wear behaviour of metal gears, where the wear is zero in the pitch point. It is also the fact in our model that the wear depth of model 2 is overall higher. Namely, model 1 calculates wear over the number of cycles progression linearly, while model 2 calculates wear according to previous wear. Furthermore, some explanations for Fig. 9 are added in the revised manuscript.

Comment #5: In the conclusion part, authors reported that “Due to the contact problem and the formation of contact pressure peaks in the finite element mesh, it is necessary to consider mesh smoothing in the model to allow a smooth distribution of wear over the surface. This avoids additional convergence problems in the use of BDM, but it does result in the averaging of values in locations where the differences in wear between adjacent element nodes should actually be larger (such as pitch point) and is not entirely correct.” Could authors simulate 3 different areas including the first point of contact, pitch point, and last point of contact with 3 different constant mesh updating?

Response:

In our model, the mesh updating depth is computed in a general way independent of the application (gear wear analysis in only one of the applications) and on a 3D surface. So, for a general model, it is very hard to determine where the averaging is useful and where it will impact the results in an unwanted manner. Maybe a different averaging technique could be used, but it would have to be compared to experimental data, which is not currently available.

Comment #6: Is it possible to use both tested gear and supported gears made by POM? The existing Archard's wear model could include incorporating a contact stress-dependent wear coefficient. If authors consider the contact stress-dependent wear, is it necessary to use both tested gear and supported gears made by POM?

Response:

Yes, the modelling of both gears made from POM is possible. The stress-dependent wear coefficient could be added to the model, but the analysis results must be compared to the experimental data to set the stress dependence properly. If a stress-dependent wear coefficient would be used, it would not impact the gear material selection. All combinations of gear materials could be analysed.

Comments on the Quality of English Language

Comment #: The grammar errors and mistyping words were found. Please check carefully the manuscript again.

Response: The revised version of the manuscript has been examined by Mrs. Shelagh Margaret, our English language proofreader.

Reviewer 3 Report

Comments and Suggestions for Authors

Overall comments:

The study provides a computational model for evaluating the wear behaviour of polymeric gear components. Although the developed model provides a significant advance in the field of meodelling wear behaviour of materials, the scenario studied in the model is very limited. The model should be applied to different conditions (i.e., different gear types and designs, different wear conditions, contact pressure, materials, etc.), and then the estimated results should be compared and correlated with analytical or experimental results. Thus, the paper should be extended as suggested and resubmitted.

Specific comments:

Title: 

The title should be specified in a way that highlights the exact application of the model (e.g. dry sliding wear ....) and material (e.g., types of polymers).

Abstract:

Again, the intended wear application should be better described! The authors should emphasise the exact wear type and conditions under which the developed model can be implemented. 

Methods and materials should be further described.

Some example engineering fields for the use of polymer gears should be underlined.

Introduction:

A literature discussion including the most relevant and recent papers should be given. Then, the originality of the present work should be highlighted. 

Materials and Methods:

Please enhance the resolution of Figure 5.

Results and discussion:

As given in the overall comments, the study should include some more results in terms of the implementation of the developed model. Then, any discrepancies should be described, and the limitations should be discussed. 

Conclusions:

Upon the limited results presented, I believe that it is not possible to derive such conclusions: "The proposed computational model could be used to simulate the wear behaviour of 355 contacting mechanical elements like gears, bearings, etc." Thus, the study should be extended, and then the wear conditions, types, materials, etc. for which the model can be implemented should be clarified. 

Best wishes,

Reviewer

Comments on the Quality of English Language

The quality of English language is moderate. 

Author Response

Overal comments

The study provides a computational model for evaluating the wear behaviour of polymeric gear components. Although the developed model provides a significant advance in the field of meodelling wear behaviour of materials, the scenario studied in the model is very limited. The model should be applied to different conditions (i.e., different gear types and designs, different wear conditions, contact pressure, materials, etc.), and then the estimated results should be compared and correlated with analytical or experimental results. Thus, the paper should be extended as suggested and resubmitted.

Response: As explained in the Conclusions of the revised manuscript, the proposed computational model could be extended considering different conditions, such as different gear designs, different wear conditions, different materials, etc.).

Title

The title should be specified in a way that highlights the exact application of the model (e.g. dry sliding wear ....) and material (e.g., types of polymers).

Response: The Title of the revised manuscript was changed as suggested. The new Title is now: “A computational model for analysing the dry rolling/sliding wear behaviour of polymer gears made of POM”

Abstract

Again, the intended wear application should be better described! The authors should emphasise the exact wear type and conditions under which the developed model can be implemented. Methods and materials should be further described. Some example engineering fields for the use of polymer gears should be underlined.

Response:
In the revised manuscript, the abstract and introduction were rewritten as suggested.

Introduction

A literature discussion including the most relevant and recent papers should be given. Then, the originality of the present work should be highlighted.

Response:

The introduction was rewritten as suggested.

Materials and Methods

Please enhance the resolution of Figure 5.

Response: The resolution of Figure 5 was enhanced as suggested.

Results and discussions

As given in the overall comments, the study should include some more results in terms of the implementation of the developed model. Then, any discrepancies should be described, and the limitations should be discussed.

Response:

To highlight the relevance and limitations of the model, an analysis of the impact of smoothing and convergence analysis of the increment of cycles have been added. The model was developed and verified with comparison to the VDI analytical model. To include some more results, we would first like to validate our model with the experimental data. After the model is validated, we can apply it to other applications and scenarios.

Conclusions

Upon the limited results presented, I believe that it is not possible to derive such conclusions: "The proposed computational model could be used to simulate the wear behaviour of contacting mechanical elements like gears, bearings, etc." Thus, the study should be extended, and then the wear conditions, types, materials, etc. for which the model can be implemented should be clarified.

Response:

The proposed computational model offers a new approach to analyse the wear behaviour between gear flanks. The authors agree with the Reviewer that the model should be improved for the general investigation of wear analyses on different machine elements. In this regard, the following paragraph has been added in Conclusions:

In further work, the proposed computational model could also be extended to consider different operating conditions, such as different gear designs, wear conditions, materials, etc. Furthermore, extensive experimental investigations should be proposed to confirm computational results.

Round 2

Reviewer 2 Report

Comments and Suggestions for Authors

Thank you very much for taking the effort to revise your manuscript.